# Context-Dependent Regulation of Gene Expression by Non-Canonical Small RNAs

**DOI:** 10.3390/ncrna8030029

**Published:** 2022-04-29

**Authors:** Kinga Plawgo, Katarzyna Dorota Raczynska

**Affiliations:** 1Department of Gene Expression, Institute of Molecular Biology and Biotechnology, Faculty of Biology, Adam Mickiewicz University, 61-614 Poznan, Poland; kinpla@st.amu.edu.pl; 2Center for Advanced Technology, Adam Mickiewicz University, 61-614 Poznan, Poland

**Keywords:** noncoding RNAs, gene expression, human diseases

## Abstract

In recent functional genomics studies, a large number of non-coding RNAs have been identified. It has become increasingly apparent that noncoding RNAs are crucial players in a wide range of cellular and physiological functions. They have been shown to modulate gene expression on different levels, including transcription, post-transcriptional processing, and translation. This review aims to highlight the diverse mechanisms of the regulation of gene expression by small noncoding RNAs in different conditions and different types of human cells. For this purpose, various cellular functions of microRNAs (miRNAs), circular RNAs (circRNAs), snoRNA-derived small RNAs (sdRNAs) and tRNA-derived fragments (tRFs) will be exemplified, with particular emphasis on the diversity of their occurrence and on the effects on gene expression in different stress conditions and diseased cell types. The synthesis and effect on gene expression of these noncoding RNAs varies in different cell types and may depend on environmental conditions such as different stresses. Moreover, noncoding RNAs play important roles in many diseases, including cancer, neurodegenerative disorders, and viral infections.

## 1. Introduction

The human genome encodes only ~20,000 protein-coding genes, representing <2% of the total genome sequence. Most of the human genome consists of noncoding fragments, which have historically been regarded as junk DNA. However, the development of high-throughput technologies, such as next-generation sequencing, have allowed an in-depth examination of the noncoding genome, which revealed that the majority of genomic DNA is transcribed under some conditions. Indeed, most of the human genome is transcribed into noncoding RNAs. Moreover, some protein-coding loci also generate noncoding RNAs by alternative splicing, and in some cases the major transcript does not code a protein. Additionally, many RNAs, usually regarded as strictly protein-coding mRNAs, may also fulfill non-coding functions. Evidence from whole-genome and transcriptome sequencing suggests that the complexity of an organism may be regulated by noncoding RNAs. Consequently, the central dogma ‘DNA→RNA→protein’, which proposes that genetic information is stored in protein-coding genes, has been undermined. RNA is no longer considered to be merely an intermediary between genes and proteins. It has become increasingly apparent that noncoding RNAs are crucial players in a variety of cellular and physiological functions [1,2,3,4].

Noncoding RNAs can generally be divided into two groups: housekeeping RNAs and regulatory RNAs (Figure 1). Housekeeping RNAs are abundantly and ubiquitously expressed in cells, as they primarily regulate basic cellular functions. They include ribosomal RNAs (rRNAs) and transfer RNAs (tRNAs), which are essential elements of the translation machinery, as well as small nuclear RNAs (snRNAs) and small nucleolar RNAs (snoRNAs) implicated in diverse functions, including splicing regulation and rRNA modifications. Regulatory RNAs function as regulators of gene expression at epigenetic, transcriptional, and post-transcriptional levels. Based on their size, regulatory RNAs can be divided into two groups: small noncoding RNAs (sncRNAs), which are shorter than 200 nt, and long noncoding RNAs (lncRNAs), longer than 200 nt. Regulatory small noncoding RNAs are important regulators of gene expression in diverse biological contexts. They include, among others, microRNAs (miRNAs), small interfering RNAs (siRNAs), Piwi-associated RNAs (piRNAs), and circular RNAs (circRNAs). Furthermore, tRNAs and snoRNAs are processed into smaller fragments: tRNA-derived fragments (tRFs) and snoRNA-derived small RNAs (sdRNAs), which also have regulatory functions [5,6,7,8].

This review aims to highlight the diverse mechanisms of the regulation of gene expression by small noncoding RNAs in different conditions and different types of human cells. For this purpose, various cellular functions of miRNAs, circRNAs, sdRNAs and tRFs will be exemplified, with particular emphasis on the diversity of their occurrence and on the effects on gene expression in different stress conditions and diseased cell types.

## 2. microRNAs

MicroRNAs (miRNAs) are small, evolutionary conserved, noncoding RNAs, with an average length of 22 nucleotides. miRNAs are involved in many physiological processes, such as differentiation, proliferation, apoptosis, and development. It is estimated that 60% of human mRNAs are modulated by miRNAs [9].

About half of miRNAs are intragenic and therefore are derived from introns of protein-coding genes. The remaining miRNAs are intergenic, derived from independent transcriptional units [10]. Biogenesis of miRNAs occurs in a two-step process requiring a nuclear and cytoplasmic cleavage event (Figure 2). Firstly, primary miRNAs (pri-miRNAs) are transcribed by RNA polymerases II (Pol II) or III (Pol III). pri-miRNAs are processed into hairpin-shaped pre-miRNAs by the nuclear “microprocessor” complex, containing the RNase III endonuclease Drosha and its interacting partner DGCR8. A subset of non-canonical miRNAs called mirtrons are produced in a splicing-dependent and Drosha-independent manner. Thus, they bypass Drosha cleavage and instead depend on splicing to produce pre-miRNA molecules. The pre-miRNA is then exported to the cytoplasm by exportin-5 and is further processed by the RNase III endonuclease Dicer, which removes the terminal loop generating a miRNA duplex. This duplex is loaded onto the Argonaute (Ago) protein. Argonaute proteins are highly conserved proteins that contain domains responsible for both binding of small noncoding RNAs and cleavage of RNA substrates, and therefore constitute a main component of the RNA-induced silencing complex (RISC). At this point the passenger strand is removed or degraded, while the other is deemed the guide strand. The selection depends on the thermodynamic properties of the miRNA [11,12,13].

The guide strand and Ago form the minimal RISC (miRISC). miRNA guides miRISC to specifically recognize mRNA and post-transcriptionally regulate gene expression. The recognition occurs by the miRNA seed region, bases 2–7 or 8 of miRNA, interacting through base-pairing with complementary sequences on target mRNA, called miRNA response elements (MREs). Most miRNA binding sites lie at the 3′ untranslated region (UTR) of the target mRNA. However, miRNA binding sites have also been detected in 5′ UTR sequences, as well as in coding regions, and within promoter regions. The degree of MRE complementarity determines the gene silencing mechanism. A near-perfect base-pairing enables the Ago2 endonuclease cleavage, whereas a central mismatch induces translation inhibition [11,13].

miRNA-directed mRNA cleavage induced by a high degree of sequence complementarity is catalyzed by Ago2. However, a central mismatch in the base-pairing prevents Ago2 endonuclease activity but initiates the recruitment of proteins promoting mRNA decay through deadenylation, decapping, and exonucleolytic digestion [14,15]. Moreover, miRISC can inhibit translation initiation by releasing the eukaryotic initiation factors 4A (eIF4AI and eIF4aII) from mRNA, which impedes the assembly of the eIF4F translation initiation complex [16]. Another mechanism of miRNA-mediated translation repression is the sequestering of mRNAs away from translational machinery into cytoplasmic processing bodies (P-bodies), which are the functional site of miRNA-mediated gene silencing. P-bodies lack any translational machinery and therefore are not involved in the translation process [14,15].

Mechanisms involving miRNAs can not only inhibit but also increase translation rates of proteins. miRNA-directed translation activation can depend on the cell cycle state and proteins bound to the Ago2-miRNA complex. In serum-starved cells, the Ago2-miRNA complex associated with AU-rich elements at the 3′ UTR recruited the FXR1 (Fragile-X-mental retardation related protein 1) and activated translation [17]. Several miRNAs associate with Ago2 and FXR1 to activate translation during cell cycle arrest, but they inhibit translation in proliferating cells [10]. Specific mRNAs are also translationally upregulated in quiescent cells, such as oocytes, by an FXR1a-associated miRNA-protein complex [18]. Furthermore, miRNAs that bind to the 5′ UTR can also enhance protein synthesis. For example, miR-10a interacts with the 5′ UTR of mRNAs encoding ribosomal proteins and enhances their translation during amino acid starvation [19].

Most of the miRNA-mediated regulation of gene expression occurs in the cytoplasm and the P-bodies. However, miRNAs and Ago have also been discovered in the nucleus of human cells [20]. Human promoters contain miRNA-seed matching sites, which suggests that miRNA-mediated transcription regulation may be a common phenomenon [21]. An interaction between the promoter and miRNA may either suppress or activate transcription depending on the location of the target region and epigenetic status of the promoter [22], which is called transcriptional gene silencing (TGS) or transcriptional gene activation (TGA), respectively (Table 1). As an example, Ago2 and let-7 are involved in transcriptional repression of proliferation-promoting genes in senescence [23]. Moreover, nuclear miR-522 suppresses transcription of *CYP2E1* gene by interacting with its promoter forming a DNA:RNA hybrid to prevent Pol II and transcription factor binding to the promoter [24]. miR-223 also forms a DNA:RNA hybrid by targeting the *NFI-A* promoter region containing miR-223 complementary sequences during human granulopoiesis. It represses transcription of the *NFI-A* gene, which is an important step of granulocytic differentiation [25]. On the other hand, several miRNAs including let-7i, miR-138, miR-92a, and miR-181d bind to the TATA-box and enhance the promoter activities of interleukin-2, insulin, calcitonin, or c-myc, respectively [26]. Additionally, miRNA-373 activates transcription of E-cadherin and CSDC2 genes through enrichment of Pol II at their promoters [27]. Moreover, a subset of miRNAs deriving from the enhancer loci can induce expression of neighboring genes and function as enhancer regulators. For example, miR-24-1 increases expression of neighboring genes *FBP1* and *FANCC* by targeting enhancers. Increased miR-24-1 expression causes direct chromatin state alteration of the *FBP1* enhancer, which activates transcription. Interestingly, when miR-24-1 is located in the cytoplasm it would still be able to function as a repressor for its target genes. Thus, miRNA can carry a dual function ability: activation in the nucleus and repression in the cytoplasm [28].

Furthermore, miRNAs are also involved in the nuclear splicing program. In one mechanism, they can indirectly modulate alternative splicing by regulating translation of various splicing factors. In another pathway, Ago-miRNA complexes can regulate alternative splicing directly in the nucleus by epigenetic and non-epigenetic mechanisms. Ago2 and miRNAs binding sites were identified in the intronic sequences in human brain samples and human myocardial cells [29,30]. The proposed mechanism involves miRNA-mediated compaction of chromatin structure at specific intron-exon junctions, which slows the rate of Pol II elongation favoring exon inclusion [31].

### 2.1. miRNAs in Stress Conditions

miRNAs play key roles in mediating cellular stress responses to pathophysiological and physiological conditions, including oxidative stress, DNA damage, and oncogenic stress. miRNA biogenesis and regulating networks may be disrupted by stressors and the resulting changes in miRNA levels may induce significant physiological effects by regulating transcription factors and other signaling molecules [32,33]. An example of this mechanism is the DNA damage response. The *p53* tumor suppressor gene is a key regulator of the miR-34 family genes [34]. p53 expression level is regulated by its ubiquitin-mediated degradation and repression by miR-125b [35]. A wide range of stresses, including DNA double-strand breaks, activate p53 to arrest the cell cycle for DNA repair. Active p53 induces the transcription of miR-34a and miR-34b/c genes, which repress the expression of target genes to promote the induction of apoptosis, cell cycle arrest, and senescence [36]. Therefore, miR-34a has a complex effect on the p53 response. Moreover, miR-34a targets TP53 and MDM4, a strong p53 transactivation inhibitor, and four other post-translational inhibitors of p53. In HCT116 cells, miR-34a overexpression increases p53 protein levels and stability. However, the p53-mediated response to genotoxic stress is unimpaired in HCT116 and MCF7 miR-34a knock-out cells. The complex functional relationship between miR-34a and p53 suggests that miR-34a might act at a systems level to stabilize the p53 response to genotoxic stress [37].

Furthermore, stress conditions affecting the endoplasmic reticulum (ER) can lead to an imbalance between the demand for protein folding and the capacity of the ER protein folding, causing ER stress. Due to the accumulation of misfolded or unfolded proteins, affected cells launch the unfolded protein response (UPR). This intracellular signaling mechanism involving miRNAs can have either pro-adaptive or pro-apoptotic roles. X-box binding protein 1 (XBP1) is a pro-adaptive transcription factor that enhances secretory capacity and promotes cell survival in the adaptive UPR [32]. In a prolonged UPR, ER stress-induced apoptosis is initiated [38]. miR-30c-2-3p is upregulated during UPR and targets the 3′ UTR of *XBP1* mRNA, thereby limiting the expression of XBP1 and the survival of cells experiencing ER stress. By limiting the induction of *XBP1* mRNA, XBP1 protein, and XBP1-dependent target genes, miR-30c-2-3p could contribute to the balance between pro- and maladaptive outcomes in the UPR, considering the possible deleterious effects of excessive XBP1 expression [39]. Furthermore, several miRNAs influence apoptotic signaling pathways in ER stress. miR-211-5p and miR-204-5p regulate the expression of several ER stress markers, particularly the pro-apoptotic transcription factor CHOP. Blocking of endogenous miRNA-211-5p and miR-204-5p increases human beta cell apoptosis [40]. Furthermore, miR-204 was implicated in ER stress-responsive gene modulation and apoptosis susceptibility in human trabecular meshwork cells. miR-204 inhibited the induction of ER-stress response markers. Overexpression of miR-204 increased apoptosis and cell death in response to oxidative stress and ER stress [41]. miR-96-5p, another miRNA involved in the response to ER stress, is involved in the non-cholinergic toxicity of malathion in normal human kidney cells (HK-2 cells). miR-96-5p protects HK-2 cells from malathion-induced ER stress-dependent apoptosis by targeting DDIT3, a well-known marker of ER stress [42]. 

### 2.2. miRNAs in Cancer

Cancer progression is a multistep process including mutation and selection for cells with increasingly enhanced proliferative, survival, invasion, and metastatic capacities. Cancer cells have several distinct characteristics enabling their autonomous excessive proliferation. They can proliferate independently of growth signals and are unresponsive to inhibitory growth signals, resulting in limitless replicative potential. Furthermore, cancer cells evade apoptosis, induce and sustain angiogenesis, and form new colonies discontinuous with the primary tumor [43].

All tumors present specific signatures of altered miRNAs expression. Generally, downregulation of miRNA expression is observed in many cancers, although upregulated expression of some miRNAs can also be directly correlated with tumor development. In addition to the distinction of tumors from normal tissues, miRNA expression is characteristic for a cancer type, stage, and other clinical variables [44]. Different miRNAs can act as tumor suppressor genes and/or oncogenes (Table 2).

let-7 is one of the earliest discovered miRNAs. It is considered a tumor suppressor gene partly due to its normal physiological role in arresting cell development [45]. While let-7 is almost absent during embryonic stages or tissues, it is highly expressed in most differentiated tissues. The reduction of let-7 in cancers is similar to let-7 expression during development, as its expression is lowest in less differentiated, advanced stages of cancer [60]. let-7 targets transcripts critical for DNA replication such as *PD-L1* (programmed cell death ligand 1) and *HMGA2* (high mobility group AT-hook 2), as well as apoptotic genes [45]. Moreover, the miR15/16 cluster acts as a tumor suppressor by targeting *BCL2* mRNA. The oncogenic BCL2 protein, commonly overexpressed in various cancers, promotes cell survival by evading apoptosis. In normal conditions, miR-15/16 directly targets and negatively regulates *BCL2*, inducing intrinsic apoptosis pathways [46]. However, the miR-15a/16-1 cluster is a frequently deleted region in B cell chronic lymphocytic leukemia (CLL) and other cancers, resulting in higher expression of BCL2 oncogene [61].

On the contrary, some miRNAs have been found to function as oncogenes. miR-21 downregulates four tumor suppressors: mapsin, PDCD4 (programmed cell death 4), TMP1 (tropomyosin1), and PTEN (phosphatase and tensin homolog). miR-21 binds to the 3′ UTR of the gene transcripts, preventing their translation. This in turn promotes cell transformation, tumor growth, invasion, and metastasis [55]. miR-21 overexpression was observed in a variety of cancers including lung, breast, and bladder cancer [47,48,49]. miR-155 is also known to be oncogenic in multiple tumors. Several signaling pathways, such as TGF-β and JAK-STAT, are under the control of miR-155 [50]. In breast cancer cells, miR-155 was shown to enhance tumor growth, promote cell proliferation and inhibit apoptosis [51]. Stimulation of breast cancer cells by inflammatory cytokines significantly upregulates miR-155 expression, suggesting that miR-155 may serve as a bridge between inflammation and cancer [62].

Because of the miRNA expression patterns, which can differ for specific tissues and differentiation states, and the fact that a single miRNA can regulate multiple targets, some miRNAs can function as both tumor suppressors and oncogenes in different contexts. miRNA-125b is upregulated in some tumor types, such as colon cancer and hematopoietic tumors, and displays oncogenic potential, as it induces cell growth and proliferation while blocking the apoptotic machinery. However, in other tumor entities, including non-small cell lung cancer (NSCLC) and breast cancer, miRNA-125b is heavily downregulated, which is accompanied by de-repression of cellular proliferation and anti-apoptotic programs, contributing to malignant transformation. These opposing roles might be explained in that miR-125b targets multiple mRNAs, which have diverse functions in individual tissues. miR-125b is regulated by multiple factors, and the interaction of miR-125b with its targets constitutes their regulatory network. Therefore, because of the complexity of this network, miRNA-125b has a wide range of cellular functions, depending on the context [52,53,54].

Similarly, the miR-17-92 cluster also has a dual role as both an oncogene and tumor suppressor. This cluster is highly elevated in a variety of lymphomas, as well as lung, colon, pancreas, and prostate cancers [55]. The miR-17-92 cluster constitutes a very complex regulatory network with c-myc and E2F transcription factors, which are critical regulatory components for apoptosis and cell proliferation. miR-17-5p and miR-20a of the miR-17-92 cluster inhibit the translation of *E2F* mRNA [56]. Furthermore, miR-17-5p acts as a tumor promoter and prognostic factor for recurrence in head and neck squamous cell carcinoma (HNSCC) [57]. However, in some cases, the miR-17-92 cluster was found to act as a tumor suppressor. miR-17-5p downregulates the proto-oncogenic transcriptional activator AIB1 (amplified in breast cancer 1), which is involved in breast cancer proliferation, growth, and hormone signaling [55]. miR-17-5p also acts as a tumor suppressor by targeting the ETV1 (ETS variant 1) transcription factor, which promotes triple-negative breast tumor cell proliferation, invasion, and migration. The expression of miR-17-5p in triple-negative breast cancer (TNBC) cell lines reduced the expression of ETV1 and consequently inhibited cell proliferation, migration, and invasion, which might suppress the development of TNBC. High miR-17-5p expression was associated with a significantly favorable prognosis [58]. Moreover, miR-17-5p and miR-20a-5p from the miR-17-92 cluster play inhibitory roles on hepatocellular carcinoma (HCC) metastasis. miR-17-5p and miR-20a-5p both target the 3′ UTR of the oncogene ERBB3. They could suppress postoperative metastasis of HCC [59].

Aside from the miRNA-mediated post-transcriptional regulation, nuclear miRNAs, which mainly act upon the promoter region, can regulate the transcription of tumor-suppressor genes, oncogenes, or other cancer-related genes. The miR-877-3p binding site was found on the promoter of tumor suppressor gene *p16*. Enforced expression of miR-877-3p can increase the expression of p16, which inhibits proliferation and tumorigenicity of bladder cancer through cell cycle G-1 phase arrest [63]. In turn, miR-6734 inhibits the growth of colon cancer cells by upregulating *p21* gene expression, which leads to induction of cell cycle arrest and apoptosis [64]. On the other hand, miR-483 is an oncogenic intronic miRNA that binds to *IGF2* (imprinted insulin-like growth factor 2) gene promoter. Ectopic expression of miR-483 induces upregulation of IGF2 expression, as well as an increase in tumor cell proliferation, migration, invasion, and tumor colony formation in HCC [65].

### 2.3. miRNAs in Viral Infections

Severe acute respiratory syndrome coronavirus 2 (SARS-CoV-2) emerged in December 2019 in China and caused the global pandemic of coronavirus disease 2019 (COVID-19) [66]. The plasma miRNA profile is deeply disrupted by SARS-CoV-2 infection, making miRNAs highly valuable as early prognostic biomarkers of severity and mortality [67]. The differential miRNA expression in COVID-19 patients may regulate the immune responses and viral replication during infection. COVID-19 exploits the interplay between miRNA and other biomolecules to avoid being effectively recognized and attacked by the host’s immune system. SARS-CoV-2 adsorbs host immune-related miRNAs and participates in the maladjustment of the host’s immune systems. The virus also encodes its own miRNAs, which can enter the host cell and are not perceived by the host’s immune system [68,69,70].

Viral miRNAs can either interact with specific regions of their own genome or transcript or bind to host miRNAs and genes during virus infection [70]. Six SARS-CoV-2 miRNAs were predicted to interact with human miRNAs targeting immune genes and result in cytokine storm, characterized by the excessive release of pro-inflammatory cytokines [71,72]. SARS-CoV-2 miRNAs have been shown to inhibit ribosomal translation of some key human proteins by hybridizing their mRNAs. This inhibition concerns, for example, the hemoglobin and type I interferon protein synthesis, hence highly perturbing oxygen distribution in vital organs and immune response in COVID-19 patients [73]. Furthermore, a virus-encoded miRNA MR147-3p can enhance the *TMPRSS2* gene expression in the gut. TMPRSS2 is involved in the activation of the spike protein of the virus binding to the ACE2 (angiotensin-converting enzyme 2) receptor. Consequently, the virus-derived MR147-3p facilitates effective evasion of the virus into gut cells [74]. Host miRNAs might block attachment and entry of SARS-CoV-2 because several miRNA families target *ACE2* and *TMPRSS2* genes in different tissues. For instance, miR-200c suppresses *ACE2* mRNA and protein expression in cardiomyocytes [75]. hsa-miR-98-5p targets *TMPRSS2* 3′ UTR inhibiting transcription of *TMPRSS2* gene in human lung microvascular endothelial cells and human umbilical vein endothelial cells [76]. In addition, many human miRNAs target and inhibit viral genes involved in replication, translation, and protein synthesis by interacting with MREs within the 5′ UTRs and 3′ UTRs of the viral genome [77]. For instance, miR-1307-3p and miR-3613-5p were predicted to prevent virus replication by binding to 3′ UTRs of its genome [78].

However, SARS-CoV-2 infection-induced host miRNAs can also function as pro-viral factors by targeting several important host immune surveillance pathways [79,80]. One of the pathways downregulated by host-miRNAs during SARS-CoV-2 infection is the signaling of different toll-like receptors (TLRs), which are the primary stimulatory molecules for producing host antiviral responses. Other receptor signaling involved in antiviral responses, including uPA-UPAR signaling, TRAF6 signaling, S1P1 signaling, Estrogen receptor signaling, protease-activated receptor (PAR) signaling, and bone morphogenetic protein (BMP) signaling, can also be deregulated by the host miRNAs, leading to the host’s immune suppression [69]. 

Alterations in the expression of miR-200c-3p and miR-421-5p may influence the outcomes of COVID-19 infection. These miRNAs are considered to modulate the expression of the *ACE2* gene. The expression levels of miR-200c-3p and miR-421-5p were significantly decreased, while inflammatory cytokine IL-6 expression was enhanced in COVID-19 patients at the time of admission. Targeting miR-200c-3p and miR-421-5p can maintain the level of ACE2 and modulate the inflammation [81].

### 2.4. miRNAs in Neurological Disorders

miRNAs are important regulators of intracellular pathways for various neurodegenerative conditions. Altered levels of numerous miRNAs were found in several pathological conditions including Alzheimer’s disease (AD) and Parkinson’s disease (PD) [82].

Alzheimer’s disease (AD) is the most common age-related neurodegenerative disorder. In AD, the accumulation of amyloid β peptides in plaques and the presence of phosphorylated Tau aggregates in neurofibrillary tangles are observed. miRNAs were found to regulate the expression levels of amyloid beta precursor protein (APP) [83]. miR-106a, miR-520c, miR-20a, miR-17-5p, and miR-106b bind to the 3′ UTR of *APP* mRNA to inhibit its translation, and thus reducing APP protein levels [84,85]. On the contrary, miR-346 promotes APP translation through interaction with its 5′ UTR. This interaction prevents the recruitment of a translation suppressor aconitase 1, indirectly stimulating APP translation [86]. Furthermore, miRNAs can regulate different steps of Tau processing. miR-15 members downregulate the mitogen-activated protein kinases 1 and 3 (MAPD1/3), which phosphorylate Tau [83]. miR-26a regulates another protein kinase, the glycogen synthetase kinase 3 beta (GSK3B), which also hyperphosphorylates Tau. GSK3B is associated with the generation of amyloid β and formation of neurofibrillary tangles in AD brains [87]. Moreover, miRNAs may be promising biomarkers in the early diagnosis of AD. For instance, the expression levels of miR-4722-5p, miR-615-3p, hsa-let7d-5p, and hsa-let7g-5p were significantly increased in AD patients. Therefore, measuring the transcript levels of these miRNAs can be used as a diagnostic biomarker for AD [88,89].

Parkinson’s disease (PD) is characterized by bradykinesia, muscle stiffness, postural instability, and involuntary tremors. These signs are related to the loss of dopaminergic neurons in the substantia nigra and the pathology spreading to other regions of the brain. The α-synuclein protein elicits the insoluble aggregates that compose the main structure of Lewy bodies leading to the death of dopaminergic neurons [90]. miR-7 inhibits α-synuclein expression directly through the 3′ UTR of α-synuclein mRNA. By downregulating the α-synuclein level, miR-7 protects cells against oxidative stress. In PD, miR-7 expression decreases, possibly contributing to increased α-synuclein expression [91]. Moreover, overactivation of microglia is one of the critical pathophysiological mechanisms underlying PD. miRNAs can affect the progress of PD by regulating the expression of various microglia genes and the polarization state of the microglia [92]. For example, miR-132-3p promotes neuroinflammation and dopaminergic neurodegeneration by enhancing the activation of microglia cells. Elevated expression of miR-132-3p, as well as decreased expression of its target GLRX, were found in PD patients and cell models. miR-132-3p-induced activation of microglia cells can be reversed by GLRX overexpression. Suppression of miR-132-3p alleviates neuroinflammation and dopaminergic neurodegeneration in PD mouse models [93].

## 3. Circular RNAs

Circular RNAs (circRNAs) are single-stranded RNAs with the 3′ and 5′ ends covalently linked. They originate from over 14% of transcribed genes in human fibroblasts. circRNAs are abundant and in some cases, they demonstrate higher expression than the associated mRNA. Large numbers of circRNAs can accumulate in the cytoplasm of cells because of their resistance to degradation by mechanisms recognizing the ends of linear RNAs. The lack of 5′–3′ polarity and a polyadenylated tail gives them a higher tolerance to endonucleases, making them highly stable, with transcript half-lives exceeding 48 h [94,95].

Circular RNAs are generally formed by alternative splicing of pre-mRNA in a process known as ‘backsplicing’, in which an upstream splice acceptor is joined to a downstream splice donor (Figure 3). Nearly all circular RNAs contain exonic sequences, usually between one and five exons. The canonical splicing machinery functions in their biogenesis as both canonical splice sites are required for exon circularization. However, the exact mechanism of circRNA biogenesis and regulatory factors involved in circularization is still unclear. Proposed models include exon skipping and intron lariat formation, the presence of inverted repeats and intron pairing, as well as interaction with RNA-binding proteins (RBPs). Based on the biogenesis features and their structural domains, circRNAs can be classified into three types: exonic circRNAs (ecircRNAs), localized predominantly in the cytoplasm, circular intronic RNAs (ciRNAs), and exon-intron circRNAs (EIcircRNAs), both found mainly in the nucleus [96,97].

circRNAs are produced in a tissue-specific and developmental stage-regulated manner. Over 50% of them are tissue-specific and their number and expression levels are higher in fetal tissue than in adult tissues. Many circRNAs are highly abundant in the brain and their levels increase during neuronal differentiation and development. Large amounts of circRNAs are accumulated in neuronal tissues with aging. Consistent with these findings, a negative correlation of global circRNA abundance and proliferation was suggested, which seems to be a general principle in human tissues. Therefore, circRNAs have higher expression levels in low-proliferating cells such as cardiomyocytes compared to the high-proliferating cells of the liver [98,99,100].

Circular RNAs can regulate gene expression at both the transcriptional and post-transcriptional levels. ciRNA and EIcircRNA are mostly localized in the nucleus, where they can act as transcriptional regulators. EIcircRNAs can regulate transcription because they retain the intronic sequences from their parental gene [101]. Two EIcircRNAs, circEIF3J and circPAIP2 can interact with U1 small nuclear ribonucleoproteins (U1 snRNPs) and promote transcription of host genes by binding to Pol II [102]. In addition, ciRNA were shown to associate with the elongation Pol II machinery to activate the transcription of their parent genes [103]. For circRNAs containing intronic sequences, transcriptional activation may be their general function, which would likely explain their nuclear localization [104]. ecircRNAs that consist of exons can play a role in alternative splicing since circularization competes with canonical splicing and thus impacts linear RNA formation to modulate the transcription of related genes. The disturbed balance between circular and linear splicing can promote aberrant transcription of oncogenes or tumor-suppressor genes in cancer. Moreover, the formation of an ecircRNAs can act as an ‘mRNA trap’ by sequestering the translation start site and leaving a noncoding linear transcript. This mechanism could be very widespread, as many single ecircRNAs in human fibroblasts contain a translation start [105].

circRNAs were found to act as protein sponges, scaffolds, decoys, and recruiters. Some of them can regulate gene expression by interacting with proteins to either promote or inhibit translation. Many circRNAs contain miRNA response elements and act as competing endogenous RNAs to restrain miRNAs from negatively regulating their target mRNAs. Thus, by acting as miRNA sponges, circRNA can indirectly regulate the translation of mRNAs [100,105].

### 3.1. circRNAs in Stress Conditions

Circ_0010729 participates in the regulation of hypoxia-induced cardiomyocyte injuries by mediating the miR-27a-3p/TRAF5 axis. Hypoxia inhibits cell viability, induces apoptosis, and blocks glycolysis. Circ_0010729 targets miR-27a-3p, which in turn upregulates the expression of the tumor necrosis factor receptor-associated factor 5 (TRAF5). Therefore, miR-27a-3p inhibition leads to the deterioration of hypoxia-induced cardiomyocyte injuries and consequently, these injuries were alleviated by circ_0010729 knockdown [106]. Moreover, the hypoxia-induced upregulation of circ-0010729 was shown to regulate vascular endothelial cell proliferation and apoptosis via targeting the miR-186/HIF1α axis. Knockdown of circ-0010729 suppressed the proliferation and migration ability and enhanced apoptosis [107].

Hsa_circ_0005915 is significantly upregulated in HL-7702 liver cells under oxidative stress. Hsa_circ_0005915 downregulates the expression of nuclear factor erythroid-2-related factor 2 (NRF2) by promoting its ubiquitination and degradation, which leads to increased accumulation of reactive oxygen species (ROS). Overexpression of has_circ_0005915 causes a decline of the expression levels of NRF2-regulated antioxidative genes: heme oxygenase 1 (*HO1*) and NAD(P)H quinone dehydrogenase 1 (*NQO1*) [108].

Circ-RBMS1 was found to be highly expressed in chronic obstructive pulmonary disease (COPD) patients. Cigarette smoke extract (CSE) increases expression of circ-RBMS1 in a dose-dependent manner. Knockdown of circ-RBMS1 alleviates CSE-induced oxidative stress, inflammation, and apoptosis. Circ-RBMS1 targets miR-197-3p, which in turn targets the F-box only protein 11 (FBXO11). FBXO11 promotes the degradation of proteins critical in regulation of the cell cycle, such as BCL6, SNAIL and p53. Therefore, by acting as a sponge for miR-197-3p, circ-RBMS1 positively regulates FBXO11 expression in bronchial epithelial cells, enhancing CSE-induced apoptosis, inflammation, and oxidative stress [109,110].

### 3.2. circRNAs in Cancer

*FLI1* exonic circular RNA (FECR1) formed by the proto-oncogene Friend leukemia virus integration 1 (*FLI1)* was identified upregulated in advanced metastatic breast cancer. FECR1 uses a positive feedback mechanism to activate FLI1 by inducing DNA hypomethylation in CpG islands of the promoter. FECR1 recruits TET1, a demethylase to the promoter, and downregulates DNMT1, a methyltransferase essential for the maintenance of DNA methylation. By regulating the DNA methylating and demethylating enzymes, FECR1 acts as an upstream regulator to control breast cancer tumor growth. Overexpression of FECR1 was shown to enhance the invasiveness of breast cancer cells [111]. Similarly, in gastric cancer (GC), circ-DONSON recruits the NURF complex to the *SOX4* promoter and initiates its transcription, thus promoting the proliferation, migration, and invasion of GC cells [112]. circAMOTL1 can interact with the proto-oncogene transcription factor c-myc increasing its retention in the nucleus and promoting its stability. Consequently, circAMOTL1 upregulates the expression of c-myc targets, including HIF-1α, Cdc25a, ELK-1, and JUN [113]. circYAP negatively regulates the yes-associated protein (Yap) translation by suppressing the assembly of Yap translation initiation machinery [114]. Yap activation can promote proliferation, inhibit apoptosis, and facilitate the metastasis of cancer cells [105].

ciRS-7 (also known as CDR1as), the circRNA sponge for miR-7, contains over 70 miR-7 binding sites and is expressed in different tissues and organs. By recruiting miR-7, ciRS-7 inhibits its function and upregulates the expression of related genes, which are central oncogenic factors in cancer-associated signaling pathways. Therefore, ciRS-7 acts as an oncogene and promotes tumor progression through competitively inhibiting miR-7 (Table 3). ciRS-7 is upregulated in several cancers [94,115]. ciRS-7 levels increase with the development of NSCLC and negatively correlate with the expression of miR-7 but positively correlate with the expression of its key target genes and consequently promote the tumor progression in NSCLC [116]. ciRS-7 is also upregulated in colorectal cancer (CRC). ciRS-7 overexpression impairs tumor suppressive effects of miR-7 in CRC cells, resulting in a more aggressive oncogenic phenotype. Overexpression of ciRS-7 was associated with poor patient survival and emerged as an important risk factor for overall survival [117].

Likewise, circHIPK3 is also upregulated in CRC and was found to sponge the tumor suppressor miR-7. Consequently, circHIPK3 has a tumor-promoting role in CRC [118]. Furthermore, elevated expression of circHIPK3 was found in gallbladder cancer (GBC) cell lines. circHIPK3 sponges the tumor-suppressive miR-124 leading to an increased expression of miR-124 targets, including ROCK1 (rho-associated protein kinase 1) and CDK6 (rho-associated protein kinase) [119]. In HCC, circHIPK3 sponges miR-124 and miR-506 to upregulate PDK2, which results in accelerated proliferation and invasion of HCC cells [120].

circITCH derives from the *ITCH* gene, which encodes a member of the E3 ubiquitin-protein ligase family and some of its targets are associated with tumor formation and cancer progression. circITCH has multiple binding sites, which were also found in the 3′ UTRs of *ITCH* transcripts. Therefore, the sponging of target miRNAs by circITCH regulates ITCH levels. The presence of circITCH in cells leads to an increase in *ITCH* mRNA levels resulting in greater ITCH ubiquitin activity, which decreases the activity of oncogenic factors [129]. In CRC, the expression of circITCH was downregulated in cancerous cells, resulting in low levels of *ITCH* gene expression. In addition, circITCH is likely involved in a signaling pathway regulating cell proliferation and migration and may have an anti-proliferative role in CRC [121]. Furthermore, circITCH plays a crucial role in suppressing lung cancer progression by functioning as a sponge for miR-7 and miR-214 [104]. The circITCH/miR-214 axis regulates nasopharyngeal carcinoma (NPC) proliferation, migration, and invasion through regulating the expression of PTEN, a direct target of miR-214. circITCH suppresses NPC tumorigenesis by upregulating PTEN expression through inhibiting miR-214. In NPC, the level of circITCH was decreased, while the level of miR-214 was increased [122].

Another circRNA, circNT5E, is an oncogene in glioma. circNT5E acts as a sponge of the brain-enriched miR-422a, which acts as a tumor suppressor by suppressing proliferation and invasion and inducing apoptosis in glioblastoma cells. circNT5E binds miR-422a and inhibits its activity, thus promoting the pathological development of human glioblastoma [123]. In NSCLC, circNT5E levels are significantly elevated, while its target miR-134 is downregulated, which promotes progression of NSCLC [124]. circNT5E was also found to be elevated in bladder cancer tissues and its expression level correlated with larger tumor size and lower survival rate. In this case, circNT5E sponges miR-502-5p to increase the expression of HOXC8, which promotes tumor growth and metastasis [125]. 

The expression of circRNAs can be induced by hypoxia, which is a key feature of the tumor microenvironment and impacts the cancer aggressiveness and therapy [130]. Hypoxia induces the expression of circDENND2A in glioma, which sponges the tumor suppressor miR-625-5p. miR-625-5p inhibits proliferation and increases the chemosensitivity of glioma cells. By sponging miR-625-5p, circDENND2A promotes the migration and invasion of glioma cells [126]. circDENND4C, highly expressed in breast cancer, is upregulated in response to hypoxia. It regulates breast cancer progression by sponging miR-200b and miR-200c. miR-200b and miR-200c serve as important tumor suppressors by inhibiting cell proliferation, migration, and invasion in breast cancer. Knockdown of circDENND4C suppresses glycolysis, migration, and invasion in breast cancer cells under hypoxia by increasing miR-220b and miR-200c [127]. circDENND4C was also found to act as an oncogene in other types of cancer. For example, in lung cancer circDENND4C upregulates BRD4 by sponging miR-141-3p, thereby promoting metastasis and proliferation of NSCLC [128]. 

### 3.3. circRNAs in Viral Infections

Viral infections can change the expression profile of circRNAs in the host cells. However, the genome of many viruses can induce circRNAs to regulate viral infections and pathogenesis. 

Altered expression of circRNAs was found during human cytomegalovirus (HCMV) latent infection. 1421 differentially expressed circRNAs were identified in latently infected human leukemia monocytes (THP-1 cells). Those circRNAs mainly targeted genes involved in the regulation of secretion pathways, cell cycle, and apoptosis. A potential target of the differentially expressed circRNAs could be hsa-miR-21, an important intrinsic antiviral factor that is regulated by HCMV infection. hsa-miR-21 plays an important role in regulating the cell cycle, controlling tumor growth, and reducing apoptosis [131].

circPSD3 is upregulated in hepatitis C virus (HCV) infected liver cells. circPSD3 displays a pro-viral effect and promotes replication of HCV genotype 1 and 2. circPSD3 sequesters eIF4A3, thereby inhibiting the cellular nonsense-mediated decay (NMD) pathway in HCV-infected liver cells and aiding in viral replication [132].

Recently, 3437 circRNAs encoded by SARS-CoV-2 were identified. In SARS-CoV-2, viral circRNAs downregulate genes associated with metabolic processes of cholesterol, alcohol, sterol, and fatty acid and upregulate genes associated with cellular responses to oxidative stress in the late stage of viral infection. Several genes regulated by viral circRNAs participate in biological processes such as response to reactive oxygen and centrosome localization [133]. Moreover, 6118 circRNAs were identified in human lung epithelial cells infected with SARS-CoV-2, including 477 novel circRNAs. These circRNAs are derived mainly from genes involved in immune and inflammatory responses. Therefore, SARS-CoV-2 infection significantly impacts the circRNA expression profiles of the host, suggesting their potential roles during virus infection. A total of 43 circRNAs were significantly dysregulated following infection at multiple phases. The differential expression of circRNAs was gradually increased with progression of infection. The dysregulated circRNAs could regulate mRNA stability, immunity, and cell death by binding specific proteins [134].

### 3.4. circRNAs in Neurological Disorders

circRNAs are specifically enriched in brain tissue. Intriguingly, they are differentially expressed in various brain regions and subcellular compartments as well as at specific embryonic and postnatal stages. circRNAs were found to be highly enriched in synapses, implicating a role for circRNAs in neuronal development and plasticity. Furthermore, circRNAs can be up- or downregulated in cultured neurons in response to fluctuations in neuronal activity, suggesting that neuronal activity, which affects gene expression in multiple ways, could potentially also do so by interfering with circRNA levels [135,136].

circRNAs may be involved in a wide range of neuronal stress responses and their aberrant expression or function may contribute to the pathogenesis and progression of neurological diseases. In AD, functional deficiency of ciRS-7 can upregulate miR-7 expression and may lead to downregulation of AD-relevant targets, such as ubiquitin protein ligase A. This autophagic protein is important for clearing amyloid peptides and is less abundant in the AD brain. Consequently, ciRS-7 may participate in AD pathogenesis [135,137,138].

Aberrant expression of circRNAs was also found in PD. In the healthy substantia nigra, circRNAs accumulate in an age-dependent manner. However, in the PD substantia nigra this correlation is lost and the number of circRNAs is reduced. In other studied brain regions of PD patients, the levels of circRNAs are increased. cirSLC8A1 increases in the substantia nigra of PD individuals and sponges miR-128. Targets of miR-128 include the neurodegeneration and aging-related *BMI1*, *SIRT1*, and *AXIN1* transcripts. Concordantly, these transcripts are increased in PD patients [136,139].

## 4. snoRNA-Derived Small RNAs

Small nucleolar RNAs (snoRNAs) are 60–300 nt noncoding RNAs that accumulate in the nucleolus. snoRNAs comprise two families, C/D and H/ACA box RNA, that function as ribonucleoprotein (RNP) complexes to guide modification and processing of other RNAs, mainly ribosomal RNAs (rRNAs). Over 400 different snoRNA species have been identified in the human genome. However, only half of them have predicted target sites. The remaining and increasing number of snoRNAs, which could have different functions, is referred to as “orphan snoRNAs”. Other functions fulfilled by snoRNAs include metabolic stress regulation and modulation of alternative splicing. Furthermore, snoRNAs undergo further processing into stable shorter fragments called snoRNA-derived RNAs (sdRNAs). sdRNA production is widespread, as over half of all snoRNAs produce smaller fragments. sdRNAs are similar in size to miRNAs as they typically vary from 20 to 30 nt. snoRNA-derived RNAs larger than 22 nt are also referred to as processed snoRNAs (psnoRNAs). The term sdRNAs describes both psnoRNAs and snoRNA-derived miRNAs [140,141,142,143].

sdRNAs associate with different proteins to snoRNAs, suggesting that they fulfill distinct cellular functions. They can regulate alternative splicing events and have miRNA-like abilities [144,145]. sdRNAs were observed in most if not all species’ genomes containing snoRNA genes with conserved processing profiles, which suggests their functional relevance [146]. Common processing patterns have been observed across snoRNAs and the processing extent of different snoRNAs may differ in the same cell line, suggesting that sdRNAs are not simply degradation products [147]. sdRNAs derived from C/D snoRNAs are predominantly ~17–19 nt and ~30 nt and originate from the 5′ end, whereas H/ACA box snoRNAs generate 20–24 nt-sized fragments mainly derived from the 3′ end [148]. Many H/ACA sdRNAs seem to follow the canonical miRNA pathway, as they are produced in a Dicer-dependent manner and are associated with Ago proteins. However, C/D box-derived sdRNAs are Dicer-independent [149]. 

Numerous C/D box-derived sdRNAs that exhibit miRNA silencing features were identified in human cell lines: HeLa, Jurkat (T cells), and RPMI8866 (B cell). Importantly, the silencing activity differed amongst all three cell types [150]. Generally, sdRNAs exhibit tissue-specific expression [141]. In addition, several snoRNA-derived miRNAs originate from orphan snoRNAs, suggesting orphan snoRNAs function as a substrate for miRNA production [140]. In turn, several known and described miRNAs genes were found to have snoRNAs precursors. Some pre-miRNAs are H/ACA-derived snoRNA, such as miR-1291/ACA34, miR-1248/HBI-6, and miR-664/ACA36b [151].

First sdRNA that was found to have miRNA-like activity is derived from the H/ACA box snoRNA ACA45. This sdRNA processing requires Dicer activity but is independent of Drosha/DGCR8. ACA45 sdRNA recognizes 3′ UTR sequences of *CDC2L6* mRNA and has the ability to post-transcriptionally silence the expression of the *CDC2L6* gene by binding to Ago1 and Ago2 proteins. Since the gene product is a component of the Mediator complex responsible for the regulation of transcription, the silencing of this gene is important for the general transcription process. Ender et al. proposed a model in which a minor portion of full-length snoRNA ACA45 is transported from the nucleus to the cytoplasm, where it is further processed to a sdRNA by Dicer [145]. 

Besides miRNA-like functions, numerous noncanonical functions, such as mediating RNA editing and splicing, have been ascribed to sdRNAs. The loss of HBII-52 and related C/D box snoRNA expression units have been implicated as a cause for the Prader-Willi syndrome (PWS). HBII-52 was found to regulate the alternative splicing of *HTR2C* (serotonin receptor 2C) pre-mRNA [152]. However, it is not the full-length cluster but sdRNAs derived from this cluster that recruit spliceosomal factors and regulate the alternative splicing of *HTR2C*. Thus, the loss of the regulatory psnoRNAs could contribute to the etiology of PWS [144]. Furthermore, over a hundred alternative splicing target sites for HBII-85 and five other orphan snoRNAs were predicted to have a significant association with alternatively spliced genes [153]. SNORD88C (HBII-180C) is another snoRNA that is processed and shows complementarity to a pre-miRNA. HBII-180C produces sdRNAs containing the M-box that is complementary to several pre-mRNAs including *FGFR3* and can regulate splicing through interactions with pre-mRNA regulatory elements [154]. The interaction between the HBII-180C M-box and an intronic *FGFR3* element leads to increased exon inclusion [155].

Recently, a new class of nucleus-localized small RNAs called snRNA/snoRNA-derived nuclear RNAs (sdnRNAs) was identified. The most abundant one, sdnRNA-3, was shown to inhibit the transcription of the *Nos2* gene by decreasing chromatin accessibility of the gene promoter in macrophages. sdnRNA-3 promotes the enrichment of the repressive chromatin-remodeling regulator Mi-2β and the repressive histone modification H3K27me3 at the *Nos2* gene promoter [156].

### 4.1. sdRNAs in Stress Conditions

The production of sdRNAs increases during stress conditions, suggesting potential roles in stress regulation. Under stress conditions, snoRNAs were shown to be significantly less abundant, while sdRNAs were significantly more abundant, suggesting stress-dependent regulation of sdRNA excision. However, snoRNA levels did not always correlate with sdRNA abundance. Interestingly, sdRNAs produced under stress conditions were found to be associated with ribosomes, suggesting yet unidentified roles of sdRNAs in translation. Considering that translation typically decreases under stress conditions, sdRNAs could potentially participate in the downregulation of protein synthesis [143,157].

### 4.2. sdRNAs in Cancer

Different cancer types are associated with unique sdRNA signatures. Even in the case of cancers from a similar tissue of origin, different cancer types exhibit divergent sdRNA expression patterns. Moreover, specific sets of sdRNAs are coordinately expressed across cancers from different tissues of origin, which suggests that sdRNA biogenesis from snoRNAs is coordinately regulated in cancer [158].

Deep sequencing of small noncoding RNAs from patients with normal prostate and prostate cancer in different stages revealed sdRNAs production from the majority of human snoRNAs. In general, sdRNAs display stronger differential expression than miRNAs and are massively upregulated in prostate cancer. At least 78 of the detected sdRNAs, including sdRNAs derived from SNORD44, SNORD78, SNORD74, and SNORD81, demonstrate strong differential expression in this cancer. Furthermore, the expression of SNORD78 and its sdRNA is significantly higher in patients that developed metastatic disease [159]. The full-length SNORD44 is encoded in one of the introns of the noncoding *GAS5* (growth arrest-specific) transcript shown to be downregulated in breast tumors [141,160]. Decreased levels of SNORD44 are associated with poor prognosis. In contrast, sdRNAs derived from SNORD44, SNORD78, and other snoRNAs encoded in *GAS5*, are upregulated in prostate cancer, which suggests separate mechanisms controlling the post-transcriptional levels of snoRNA products from the same precursor transcript [141].

The most differentially expressed sdRNA in breast cancer was found to be sdRNA-93. Increased sdRNA-93 expression in breast cancer results in enhanced invasion, while its inhibition leads to a loss of invasiveness. sdRNA-93 regulates the expression of Pipox, a sarcosine metabolism-related protein whose expression correlates with specific breast cancer subtypes and prognosis. sdRNA-93 targets *Pipox* 3′ UTR and inhibits its expression. Therefore, through participating in miRNA-like regulation of the *Pipox* gene, sdRNA-93 expression actively contributes to the malignant phenotype of breast cancer [161].

The tumor suppressor protein p53 negatively regulates transcription of the snoRNA host gene *SNHG1*, which produces several sdRNAs. The most frequently expressed sdRNA from *SNHG1*, sno-miR-28 produced from SNORD28, directly targets the p53-stabilizing gene *TAF9B*. sno-miR-28 recognizes the 3′ UTR sequence of *TAF9B* and together with the Ago protein inhibits its expression. The reduced expression in turn impairs the stability of p53. In this regulatory feedback loop p53 represses sno-miR-28 via *SNHG1* and sno-miR-28 plays an oncogenic role by targeting *TAF9B* to negatively regulate p53 stability (Figure 4a). Importantly, *SNHG1*, SNORD28, and sno-miR-28 are overexpressed in breast tumors. Moreover, overexpression of sno-miR-28 accelerates breast epithelial cell proliferation and colony formation [162]. 

Another regulatory feedback loop affecting p53 involves miR-605, the precursor of which resembles the structure of known box H/ACA snoRNAs [146,151]. miR-605 targets and post-transcriptionally represses the ubiquitin ligase MDM2, a direct inhibitor of p53. Activation of p53 upregulates miR-605 by interacting with the promoter region of the gene. Consequently, miR-605 interrupts the interaction between p53 and MDM2 to create a positive feedback loop (Figure 4b). Under stress conditions, this feedback loop could be a way for 53 to rapidly accumulate [163]. miR-605 also targets INPP4B, a phosphatase that functions as an oncogenic driver through activating SGK3 kinase in melanoma tissue. miR-605 negatively regulates INPP4B expression by repressing its mRNA translation. Thus, miR-605 functions as a tumor suppressor by inhibiting INPP4B expression and SGK3 activity in melanoma (Figure 4c). However, miR-605 is significantly downregulated in melanoma cells and tissues and therefore does not suppress the growth of melanoma cells. Generally, miR-605 functions as a tumor suppressor in various cancers [164].

Several sdRNAs are significantly correlated with features of the tumor-immune microenvironment, such as immunosuppressive markers, cytolytic T cell activity, and tumor vasculature. Many sdRNAs, including sdRNAs derived from SNORA36B in thymoma and SNORA44 in LGG, are significantly correlated with the immunosuppressive biomarker PD-L1. Various sdRNAs, including those derived from SCARNA5, SNORD6, and SNORD114-22, are strongly positively correlated with intratumoral T cell-mediated cytotoxicity by granzyme. Moreover, sdRNA expression signatures are connected to tumor vascularization. 449 sdRNAs were correlated with endothelial cell abundance in at least one cancer type. sdRNAs produced from the C/D snoRNA SNORD114-1 were found to be positively correlated with endothelial cell abundance in 16 different cancer types, suggesting that these sdRNAs play highly conserved roles in tumor vascularization across different tissues [158].

sdRNAs can also modulate cellular drug disposition. hsa-miR-1291 is a small noncoding RNA derived from SNORA34. SNORA34 was found to be processed into hsa-miR-1291 in human pancreatic carcinoma PANC-1 cells. hsa-miR01291 targets the 3′ UTR of *ABCC1* (multidrug resistance-associated protein 1) and downregulates its expression leading to a greater intracellular drug accumulation and chemosensitivity. Furthermore, hsa-miR-1291 is significantly downregulated in pancreatic ductal adenocarcinoma, compared to normal pancreas. Lower expression of hsa-miR-1291 was also found in other cancers, suggesting a common downregulation of hsa-miR-1291 in cancerous tissues [165].

## 5. tRNA-Derived Small RNAs

Transfer RNAs (tRNAs) are 76–93 nt long noncoding RNAs, essential for mRNA translation. They are the most abundant small noncoding RNAs, constituting 4–10% of all cellular RNA. During translation, tRNAs deliver amino acids to the growing polypeptide chains synthesized on ribosomes. tRNA genes are transcribed by RNA polymerase III into precursor tRNAs (pre-tRNAs), which undergo several modifications during tRNA maturation. The 5′ leader and the 3′ trailer sequences are enzymatically digested and the ‘CCA’ sequence is attached at the 3′ ends. Pre-tRNAs undergo splicing and post-transcriptional modifications. Both pre-tRNAs and mature tRNAs can be specifically cleaved into tRNA-derived small RNAs (tsRNAs), which are then modified at the 5′- and 3′-ends to stabilize them. Although tsRNAs were already discovered in the 1970s in the urine of cancer patients, it was determined only over a decade ago that they are not a degradation product, but have biological roles under different physiological and pathological conditions. tsRNAs can be divided into two main groups: tRNA-derived fragments (tRFs) and tRNA-derived stress-induced RNA (tiRNA), which are also called tRNA halves [166].

tRNA-derived fragments (tRFs) are produced from pre-tRNAs or mature tRNAs. With lengths of 14–32 nucleotides and a 5′ phosphate and a 3′ hydroxyl group, they show similarities to microRNAs. tRFs can be divided into five classes: tRF-1, tRF-2, tRF-3, tRF-5 and i-tRF (Figure 5). i-tRFs originate from the internal region of mature tRNAs and do not include the 5′-terminal and 3′-terminal regions. The exact mechanism of their biogenesis is still unclear. In addition to the above-mentioned tRFs, other tRNA fragments can also be identified by high-throughput sequencing, indicating greater diversity than existing classification [166,167,168].

Due to the structure and size similarity to miRNAs, tRFs have been hypothesized to have similar functions in inhibiting the mRNA translation. Indeed, both tRFs and miRNAs interact with Ago proteins and are involved in translation silencing. Furthermore, both are produced under stress conditions contributing to the global decrease in the translation rate during stress [170]. Some tRF-3s and tRF-5s may interact with complementary sequences on target RNAs and recruit them to Ago protein-containing complexes, which regulate the expression of these targets. The anatomy of a tRF-Target-Ago complex is similar to that of a microRNA-Target-Ago complex. tRF-3s and tRF-5s binding to Ago proteins use their 5′ seed sequence to bring the target mRNA into the Ago complexes [169]. The Argonaute-loading of tRFs is cell type-dependent, suggesting differential functional roles through the RNA interference pathway in different cell types [171].

### 5.1. tsRNAs in Stress Conditions

tRNA-derived stress-induced RNAs (tiRNAs, tRNA halves) are produced as a result of tRNA cleavage under stress conditions, such as nutrition deficiency, hypoxia, and hypothermia. Angiogenin, a stress-activated ribonuclease, cleaves tRNAs to create tiRNAs (Figure 5). The cleavage in or near the anticodon loop results in generating 5′- and 3′-fragments, which are 31-40 nt long [168].

tRNA cleavage under stress conditions was first regarded as a cellular process to decrease the mature tRNA levels. However, the pool of mature tRNAs did not significantly change with the formation of tRNA halves. Consequently, tiRNAs have an independent role in global translation inhibition [172,173]. tRNA halves are induced under oxidative stress, arsenite, heat shock, and UV radiation [174]. They inhibit translation by forming cytoplasmic stress granules (SGs) [175,176]. SGs are cytoplasmic ribonucleoprotein complexes that promote cell survival by sequestering pro-apoptotic signaling proteins while promoting the production of pro-survival proteins. Furthermore, SGs conserve anabolic energy by selectively repressing the synthesis of housekeeping proteins. There are two pathways inducing SG formation: phosphorylation of the eukaryotic translation initiation factor 2α (eIF2α) and an eIF2α phosphorylation-independent pathway [177]. Only 5′ tiRNAs, but not 3′ tiRNAs, induce the eIF2α phosphorylation-independent formation of SGs. In turn, Ala 5′ tRNA halves and Cys 5′ tRNA halves with terminal oligonucleotide motifs (four to five guanine residues) at their 5′ termini form intermolecular RNA G-quadruplexes (RG4). Then, RG4 competitively binds to translation initiation factor eIF4G in the translation initiation complex. The binding of RG4 to the translation initiation complex impairs 40S ribosome scanning on mRNA, leading to the formation of eIF2α-independent stress granules [178]. Furthermore, 5′ tRNA halves cooperate with the YB-1 protein to promote stress granules assembly. YB-1 directly binds to tiRNAs via its cold shock domain and thus enables the tiRNA-induced assembly of SGs [175].

Stress signals can also affect the stability of mitochondrial tRNAs leading to their cleavage and the production of mitochondrial tRFs (mt-tRFs). Several mt-tRFs are produced because of the excessive mitochondrial stress affecting mitochondrial DNA integrity [179].

### 5.2. tsRNAs in Cancer

tRNA halves can also be found under specific non-stress conditions, indicating that they may regulate other processes besides cellular stress responses. A type of tRNA halves, the sex hormone-dependent tRNA-derived RNA (SHOT-RNA) is induced by sex hormones, not by stress. SHOT-RNAs are produced by angiogenin-mediated anticodon cleavage promoted by sex hormones and their receptors. 5′ SHOT-RNA contains a cyclic phosphate at the 3′ end, while the 3′ terminus of the 3′ SHOT-RNA contains an amino acid, as it derives from aminoacylated tRNA. SHOT-RNAs are specifically expressed in estrogen receptor (ER)-positive breast cancer and androgen receptor (AR)-positive prostate cancer cells and are not expressed in other examined cancer cell lines [167,180].

tRF-2s derived from tRNA-Glu, tRNA-Asp, tRNA-Gly, and tRNA-Tyr are induced in hypoxic conditions prevalent in tumors and act as tumor suppressors in breast cancer. They displace the 3′ UTRs from the RNA-binding protein YBX1 that normally stabilizes oncogenic transcripts. Consequently, they cause destabilization of multiple oncogenic transcripts in breast cancer cells and contain tumor-suppressive and metastasis-suppressive activity [170]. These tRF-2s may be generated under oncogenic stress as an internal mechanism for tumor suppression. Goodarzi et. al., propose two mechanisms, possibly countering the tRF-2-mediated tumor-suppressive mechanism: avoidance of the hypoxia-evoked induction of tumor-suppressive tRFs and the upregulation of YBX1 [181]. This hypothesis is supported by findings showing decreased expression of tRF-2s in metastatic breast cancer and the upregulation of YBX1 during cancer progression [182].

tRF-3027b, also called CU1276, is a DICER1-dependent, tRNA Gly-GCC-derived tRNA fragment expressed in human mature B lymphocytes. tRF-3027b is downregulated in germinal center-derived lymphomas, suggesting a role in the pathogenesis. It associates with Ago proteins and functions as an miRNA. tRF-3027b represses endogenous *RPA1*, a gene essential for many aspects of DNA dynamics, including genome replication. Stable expression of tRF-3027b in a lymphoma cell line suppresses proliferation and modulates the molecular response to DNA damage [183].

tRF-3019a is upregulated in GC tissues and cell lines and promotes GC cell proliferation, migration, and invasion. tRF-3019a binds to the 3′ UTR of tumor suppressor gene F-box protein 47 (*FBXO47*) and interacts with Ago2 to inhibit FBXO47 expression [184].

Abnormal expression of tRFs has been found in several cancers. For example, in NPC, 158 differentially expressed tRFs were identified, of which 88 are upregulated and 70 are downregulated [185]. In CRC, the upregulation of tRF-phe-GAA-031 and tRF-VAL-TCA-002 is correlated with distant metastasis and clinical stage. These tRFs might play an important role in the metastasis of CRC [186]. Furthermore, elevated levels of i-tRF-GlyGCC were correlated with an aggressive phenotype of ovarian tumor and were linked to adverse survival outcomes [187].

### 5.3. tsRNAs in Viral Infections

tRF-3006, a tRNA-Lys-derived tRF, was found in HIV-1-infected cells. It binds to the primer-binding site (PBS) in the genomic RNA of HIV, serving as the primer for reverse transcription. The prevalence of tRF-3006 in the infected cells is positively correlated with the replication of HIV [170,188].

tRF-3019 is perfectly complementary to the PBS of the human T cell leukemia virus type 1 (HTLV-1). This tRF has been shown to prime HTLV-1 reverse transcriptase in an in vitro assay. Therefore, tRF-3019 could initiate reverse transcription and increase virus amplification [189].

tRF-5 GluCTC is induced by the human respiratory syncytial virus (RSV). Its suppression leads to a reduction in RSV viral particle production. tRF-5 GluCTC silences the apolipoprotein E receptor 2 (APOER2), which is required for activation of host immune responses. tRF-5 GluCTC binds to APOER2 3′ UTR, and thus suppresses immune responses and promotes RSV replication in RSV-infected human airway epithelial cells. Moreover, RSV leads to the induction of tRF-5 GlyCCC and tRF-5 LysCTT, which promote RSV replication and impact RSV-induced cytokines/chemokines [167].

### 5.4. tsRNAs in Neurological Disorders

In cases of neurodegenerative diseases, several mutations have been found in genes associated with tsRNA biogenesis, among them over 40 angiogenin mutants [176]. For instance, ALS-linked mutant angiogenin has limited catalytic activity and fails to induce tRNA cleavage [190]. Under stress conditions, tiRNAs (such as 5′-tiRNA Ala and 5′-tiRNA Cys) or their DNA analog inhibit protein synthesis and trigger the assembly of stress granules. They form RG4 structures, which are required for translation inhibition. The RG4 structure allows them to enter the motor neurons spontaneously and trigger a neuroprotective response [191]. However, Angiogenin can promote the accumulation of tiRNAs due to defects in tRNA methyltransferases Dnmt2 and Nsun2. The accumulation of tiRNAs triggers a stress response and cell death in the nervous system [192,193]. Therefore, tiRNAs seem to have opposite effects, either protecting neurons or promoting neuronal damage. A possible explanation could be that the roles of tiRNAs in neurons are determined by their levels and types. Production of tiRNAs at early stages could activate a stress response protecting cell survival, while in later stages of cell damage tiRNAs could lead to sustained stress and could eventually damage the cells [176].

Recently, tRFs were found to be involved in human AD. tRF-5 expression is significantly altered in the hippocampus of AD patients. Angiogenin is also enhanced in AD, suggesting its role in tRNA cleavage and tRF induction. tRF5-ProAGG is enhanced in AD and its expression is stage-dependent, suggesting its potential role as a biomarker and therapeutic target [194].

## 6. Conclusions

Noncoding RNAs regulate a large variety of cellular processes in different types of human cells. Their expression and function may depend on environmental conditions such as various stresses. Noncoding RNAs also play important roles in the pathogenesis of various diseases, including cancer, neurodegenerative disorders, and viral infections (Figure 6). Because of their involvement in pathogenesis, often in a specific manner, they may potentially serve as novel therapeutic targets. In addition, circulating noncoding RNAs could serve as diagnostic or prognostic tools, since noncoding RNA expression profiles reflect numerous pathological variables. It should be noted with respect to the tRNA fragments that they are also found in prokaryotes and appear to be involved in bacterial pathogenesis [195]. It appears that tRNA fragments function in a wide range of life forms.

The functions and detailed mechanisms of action of many noncoding RNAs are not yet completely understood. Further development of next-generation sequencing technologies and the expanding data on the expression of noncoding RNAs will clarify cellular functions and mechanistic roles of a larger spectrum of noncoding RNAs.

## Figures and Tables

**Figure 1 ncrna-08-00029-f001:**
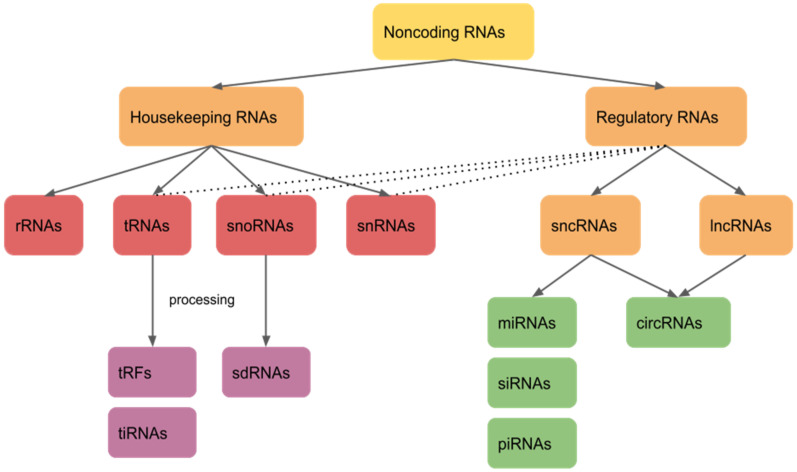
Classification of noncoding RNAs. Noncoding RNAs are divided into housekeeping RNAs and regulatory noncoding RNAs. Housekeeping RNAs include rRNAs, tRNAs, snoRNAs, and snRNAs. tRNAs and snoRNAs can be further processed into tRFs and sdRNAs, respectively. Regulatory RNAs are divided into sncRNAs and lncRNAs. sncRNAs include miRNAs, siRNAs, and piRNAs. Furthermore, circRNAs can be categorized as either sncRNAs or lncRNAs based on their size.

**Figure 2 ncrna-08-00029-f002:**
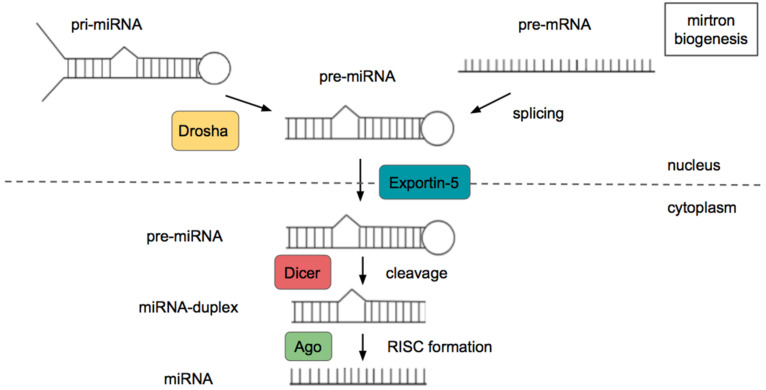
miRNA biogenesis.

**Figure 3 ncrna-08-00029-f003:**
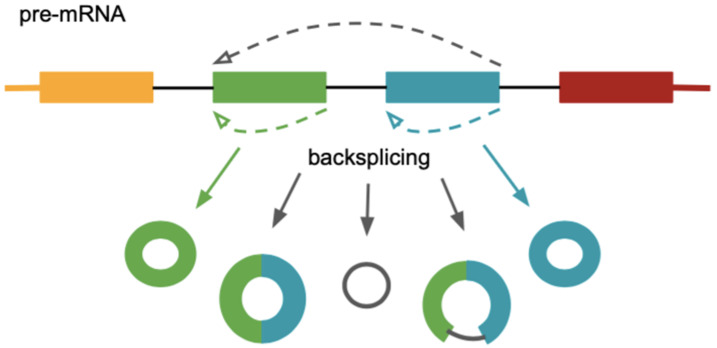
circRNAs formation by backsplicing (pre-mRNA splicing at a reversed order in which a downstream splice donor is joined to an upstream splice acceptor).

**Figure 4 ncrna-08-00029-f004:**
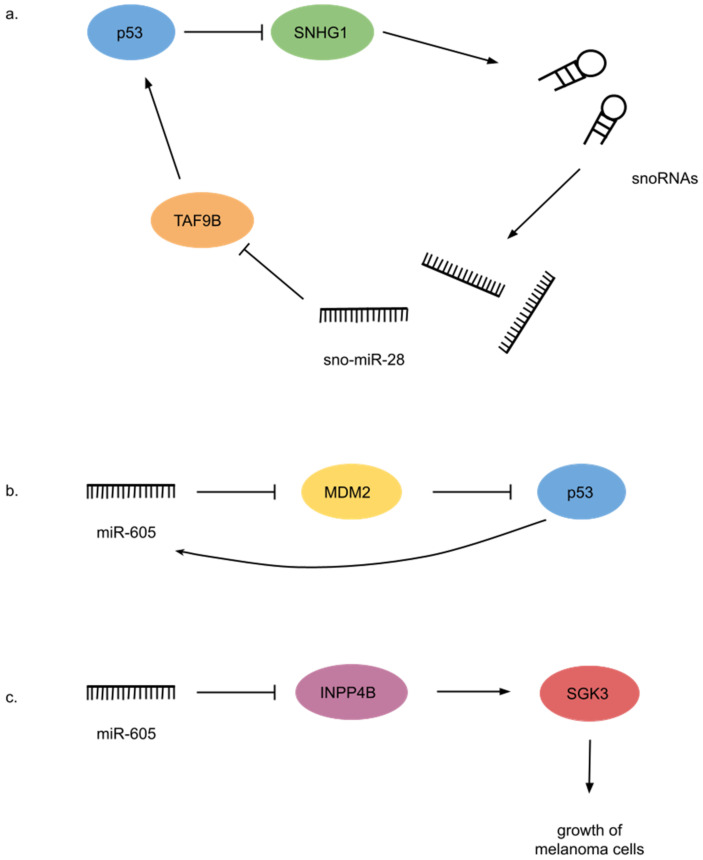
sdRNA mode of action. (**a**) p53 inhibits sno-miR-28 formation via repressing *SNGH1*. sno-miR-28 targets TAF9B to negatively regulate p53 stability. (**b**) Positive feedback loop involving miR-605, MDM2 and p53. (**c**) miR-605 represses INPP4B, and consequently SGK3, which is responsible for promoting the growth of melanoma cells.

**Figure 5 ncrna-08-00029-f005:**
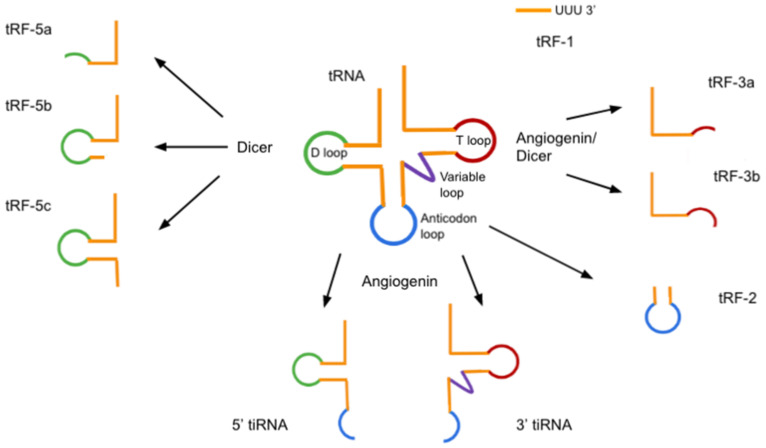
Classification of tRFs and tiRNAs. tRF-1, also called 3′ U-tRF, is generated from the 3′ untranslated region during pre-tRNA maturation and is cleaved by RNase Z or its cytoplasmic homolog ribonuclease Z 2 (ELAC2) at the 5′ end. Therefore, the 3′ end of tRF-1 contains a poly U sequence. tRF-2 contains the anticodon loop and excludes the 5′ and 3′ end structures. tRF-2s are derived from tRNA-Glu, tRNA-Asp, tRNA-Gly, and tRNA-Tyr and are induced in hypoxic conditions. tRF-3s are generated from the 3′ end of mature tRNAs by cleavage in the T-loop by angiogenin or other members of the Ribonuclease A superfamily. tRF-3s can be further divided into two subclasses: tRF3a and tRF-3b, based on their lengths of either ~18 or ~22 nucleotides, respectively. tRF-5s are produced by Dicer cutting the D-loop or the stem position between the D-loop and the anticodon loop of the mature tRNA transcript. Depending on the cleavage sites and therefore different lengths, tRF-5s are further divided into three subtypes: tRF-5a (14–16 nt), tRF-5b (22–24 nt), and tRF-5c (28–30 nt). The oligonucleotide fragments shown as stem loops, for example, tRF-2, may or may not be thermodynamically stable but are drawn as such, primarily to show the tRNA regions they are derived from [166,168,169].

**Figure 6 ncrna-08-00029-f006:**
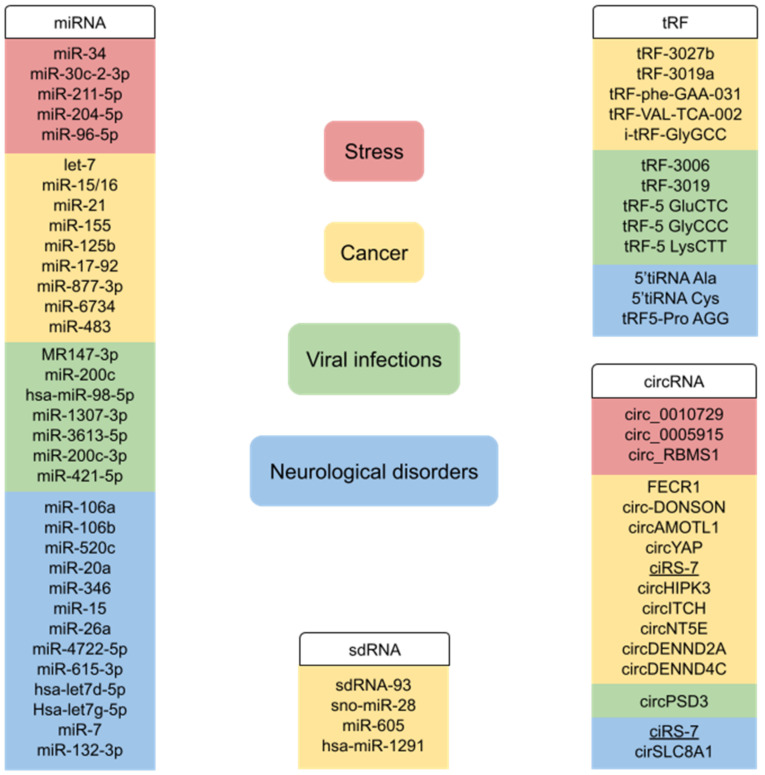
Examples of noncoding RNAs involved in stress-related processes, cancer, viral infections, and neurological disorders. RNAs involved in more than one disease are underlined.

**Table 1 ncrna-08-00029-t001:** Examples of miRNA-mediated transcription regulation in human cells.

miRNA	Function	miRNA Target	miRNA Mechanism of Action	References
miR-522	TGS	*CYP2E1* promoter	DNA:RNA hybrid with promoter	[24]
miR-223	TGS	*NFI-A* promoter	DNA:RNA hybrid with promoter	[25]
let-7i	TGA	Interleukin-*2* promoter	Binds to TATA-box and enhances promoter activities	[26]
miR-138	TGA	Insulin promoter	Binds to TATA-box and enhances promoter activities	[26]
miR-92a	TGA	Calcitonin promoter	Binds to TATA-box and enhances promoter activities	[26]
miR-181d	TGA	C-myc promoter	Binds to TATA-box and enhances promoter activities	[26]
miR-373	TGA	E-cadherin, CSDC2	Pol II enrichment at promoters	[27]
miR-24-1	TGA	FBP1, FANCC	Chromatin state alteration of the *FBP1* enhancer	[28]

TGS-transcriptional gene silencing, TGA-transcriptional gene activation.

**Table 2 ncrna-08-00029-t002:** Examples of miRNAs as tumor suppressors and oncogenes in cancer.

miRNA	miRNA Expression in Cancer	miRNA Target	Function	Type of Cancer	References
let-7	Downregulated	PD-L1, HMGA1, apoptotic genes	tumor suppressor	various cancers	[45]
miR15/16	Downregulated	BCL2	tumor suppressor	CLL	[46]
miR-21	Upregulated	Maps, PDCD4, TPM1, PTEN	oncogene	lung, breast, and bladder cancer	[47,48,49]
miR-155	Upregulated	signaling pathways (TGF-β, JAK-STAT)	oncogene	breast cancer	[50,51]
miR-125b	upregulated/downregulated	multiple mRNAs with diverse functions in different tissues	oncogene/tumor suppressor	colon cancer, hematopoietic tumors/NSCLC, breast cancer	[52,53,54]
miR-17-92	upregulated/downregulated	E2F transcription factors/AIB1, ETV1, ERBB3	oncogene/tumor suppressor	lymphoma, lung cancer, colon cancer, pancreatic cancer, prostate cancer, HNSCC/breast cancer, HCC	[55,56,57,58,59]

**Table 3 ncrna-08-00029-t003:** Examples of circRNAs acting as miRNA sponges and their functions in various types of cancer.

circRNA	miRNA Target	Disease Associated	circRNA Expression	circRNA Function	References
ciRS-7 (CDR1as)	miR-7	Several cancers including NSCLC and CRC	upregulated	oncogene	[94,115,116,117]
circHIPK3	miR-7, miR-124, miR-506	CRC, GBC, HCC	upregulated	oncogene	[118,119,120]
circITCH	miR-7, miR-214	CRC, lung cancer, NPC	downregulated	tumor suppressor	[104,121,122]
circNT5E	miR-422a, miR-134, miR-502-5p	Glioma, NSCLC, bladder cancer	upregulated	oncogene	[123,124,125]
circDENND2A	miR-625-5p	Glioma	upregulated	oncogene	[126]
circDENND4C	miR-200b, miR-200c, miR-141-3p	Breast cancer, NSCLC, lung cancer	upregulated	oncogene	[127,128]

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
