# Peer review of "Context-Dependent Regulation of Gene Expression by Non-Canonical Small RNAs"

_ncrna, 2022, doi:10.3390/ncrna8030029_

Round 1

Reviewer 1 Report

The manuscript by Plawgo and Raczyńska (ncrna-1663188) describes the recent developments and achievements made on the regulation of gene expression in four types of small non-coding RNAs (microRNAs, circular RNAs, snoRNA-derived small RNAs, and tRNA-derived fragments), in different stress conditions and diseased cell types (specifically cancer, viral infections, and neurological disorders). They provide relevant information, logically structured, regarding the basics of these ncRNAs-mediated regulation mechanisms, which are shown to be involved in important human diseases, and therefore represent a growing field of interest.

I enjoyed reading this manuscript which is well-constructed through focused sections, covering pertinent aspects of the topic and providing updated information from recent work on this topic. Further, this Review is mainly well-written, with supported conclusions, and supplemented with illustrative and helpful figures. I have some comments that could help to get an improved version:

Major comments:

  • Taking into consideration the large bibliography used for this review, I found poor the number of references utilized to sustain the Introduction chapter.
  • Related to the previous point, the definition of housekeeping and regulatory RNAs should be further explained in the Introduction (and adequately referenced), to facilitate the posterior understanding of the specific types along with the manuscript. Likewise, the term “small noncoding RNA” should be better introduced (line 45) since this Review is mainly focused on this group.
  • I consider that the following paper is relevant to be cited along the text:

Beermann et al.; Non-coding RNAs in Development and Disease: Background, Mechanisms, and Therapeutic Approaches; Physiol Rev, 2016 Oct; 96(4):1297-325; doi: 10.1152/physrev.00041.2015.

Minor comments:

  • Lines 75-76: Could that sentence be rewritten? Not sure that I understand it.
  • Line 80: The Argonaute (Ago) protein could be better introduced due to its importance along the text.
  • Line 85: is the guiding of miRNA-based miRISC towards mRNA recognition always resulting in negative regulation of the target? Please, clarify it at that point.
  • Lines 95-101: Could authors add a reference for that?
  • Line 173: Could “ER stress” be further explained?
  • Lines 196-197: could be better described?
  • Line 303: Spike protein?
  • From line 418 on: the term miRNA sponges should be introduced to improve the understanding of some parts of the text.
  • Lines 474-477: vague... please, explain it better

Author Response

Review 1

The manuscript by Plawgo and Raczyńska (ncrna-1663188) describes the recent developments and achievements made on the regulation of gene expression in four types of small non-coding RNAs (microRNAs, circular RNAs, snoRNA-derived small RNAs, and tRNA-derived fragments), in different stress conditions and diseased cell types (specifically cancer, viral infections, and neurological disorders). They provide relevant information, logically structured, regarding the basics of these ncRNAs-mediated regulation mechanisms, which are shown to be involved in important human diseases, and therefore represent a growing field of interest.

I enjoyed reading this manuscript which is well-constructed through focused sections, covering pertinent aspects of the topic and providing updated information from recent work on this topic. Further, this Review is mainly well-written, with supported conclusions, and supplemented with illustrative and helpful figures. I have some comments that could help to get an improved version:

Major comments:

  • Taking into consideration the large bibliography used for this review, I found poor the number of references utilized to sustain the Introduction chapter.

Reply – The following references were added in the introduction chapter:

  1. Ulveling, D.; Francastel, C.; Hubé, F. When One Is Better than Two: RNA with Dual Functions. Biochimie 2011, 93, 633–644, doi:10.1016/J.BIOCHI.2010.11.004.
  2. Gonzàlez-Porta, M.; Frankish, A.; Rung, J.; Harrow, J.; Brazma, A. Transcriptome Analysis of Human Tissues and Cell Lines Reveals One Dominant Transcript per Gene. Genome Biol. 2013, 14, R70, doi:10.1186/GB-2013-14-7-R70/FIGURES/5.
  3. Beermann, J.; Piccoli, M.T.; Viereck, J.; Thum, T. Non-Coding Rnas in Development and Disease: Background, Mechanisms, and Therapeutic Approaches. Physiol Rev. 2016, 96, 1297–1325, doi:10.1152/physrev.00041.2015.
  4. Zhang, P.; Wu, W.; Chen, Q.; Chen, M. Non-Coding RNAs and Their Integrated Networks. J Integr Bioinform. 2019, 16, doi:10.1515/JIB-2019-0027.
  5. Romano, G.; Veneziano, D.; Acunzo, M.; Croce, C.M. Small Non-Coding RNA and Cancer. Carcinogenesis 2017, 38, 485–491, doi:10.1093/carcin/bgx026.
  • Related to the previous point, the definition of housekeeping and regulatory RNAs should be further explained in the Introduction (and adequately referenced), to facilitate the posterior understanding of the specific types along with the manuscript. Likewise, the term “small noncoding RNA” should be better introduced (line 45) since this Review is mainly focused on this group.

Reply – Corrected as suggested. Lines 43-44; 49-50; 53-54.

  • I consider that the following paper is relevant to be cited along the text:

Beermann et al.; Non-coding RNAs in Development and Disease: Background, Mechanisms, and Therapeutic Approaches; Physiol Rev, 2016 Oct; 96(4):1297-325; doi: 10.1152/physrev.00041.2015.

Reply – The mentioned paper was cited (reference number 4).

Minor comments:

  • Lines 75-76: Could that sentence be rewritten? Not sure that I understand it.

Reply – The sentence was rewritten. Lines 83-84.

  • Line 80: The Argonaute (Ago) protein could be better introduced due to its importance along the text.

Reply – Corrected as suggested. Lines 88-90.

  • Line 85: is the guiding of miRNA-based miRISC towards mRNA recognition always resulting in negative regulation of the target? Please, clarify it at that point.

Reply – Corrected as suggested. Line 97

  • Lines 95-101: Could authors add a reference for that?

Reply – The references for the paragraph were inserted in the text, line 113 (before they were at the end of the paragraph) and the following reference was added:

  1. Wahid, F.; Shehzad, A.; Khan, T.; Kim, Y.Y. MicroRNAs: Synthesis, Mechanism, Function, and Recent Clinical Trials. Biochim Biophys Acta. 2010, 1803, 1231–1243, doi:10.1016/j.bbamcr.2010.06.013.
  • Line 173: Could “ER stress” be further explained?

Reply – Corrected as suggested. Lines 185-187.

  • Lines 196-197: could be better described?

Reply – Corrected as suggested. Lines 212-214.

  • Line 303: Spike protein?

Reply – “S protein” replaced with “spike protein”. Line 319.

  • From line 418 on: the term miRNA sponges should be introduced to improve the understanding of some parts of the text.

Reply – Corrected as suggested. Lines 436-438.

  • Lines 474-477: vague... please, explain it better

Reply – Corrected as suggested. Lines 495-498.

Reviewer 2 Report

Small noncoding RNAs  play a central role in the complex network that controls cell growth and proliferation. They have been shown to modulate gene expression on  the levels of transcription, RNA processing/decay and translation. Activity of small noncoding RNAs is  efficiently regulated, responding to a variety of external signals and cellular stress. Understanding this regulation is required for the development of new strategies for the molecular characterization and subsequent therapy of human diseases.

The review article is clearly organized. It is divided into the chapters dedicated, respectively  to miRNAs, circRNAs, sdRNAs and trfs. In the each chapter the authors  summarize  the molecular mechanisms that control expression of given small noncoding RNAs and describe their role in regulation of gene expression. Next, many examples of gene-specific regulatory effects in different cell types are described and summarized in the respective tables. Regulation became more complicated in stress conditions; it appears different for specific tissues and differentiation states. Again, many regulatory events were described in detail. That gives a broad perspective on the diversity of the regulatory effects and their significance in biogenesis of human diseases. The article is definitely worth to be published.

I have some only some minor comments concerning the text and figures

  1. The article describes regulation by small noncoding RNAs, but “small” is omitted in the title.
  2. Line 29 in the Introduction: “majority of genomic DNA is transcribed” instead “majority of all nucleotides is transcribed”.
  3. Lines 39-41: Please correct to avoid the mistakes: rRNA is not small RNA, tRNA is not constitutively expressed.
  4. Line 47: “Regulatory small noncoding RNAs….”
  5. Table 1: are there examples of miRNA-mediated regulation in human cell?
  6. Line 378: Backsplicing process is not explained and no reference is provided. Possibly the extended legend to Figure 3 would help in understanding what is going on.
  7. Figure 4 should be more compact.

Author Response

Review 2

Comments and Suggestions for Authors

Small noncoding RNAs play a central role in the complex network that controls cell growth and proliferation. They have been shown to modulate gene expression on the levels of transcription, RNA processing/decay and translation. Activity of small noncoding RNAs is efficiently regulated, responding to a variety of external signals and cellular stress. Understanding this regulation is required for the development of new strategies for the molecular characterization and subsequent therapy of human diseases.

The review article is clearly organized. It is divided into the chapters dedicated, respectively to miRNAs, circRNAs, sdRNAs and trfs. In each chapter the authors summarize the molecular mechanisms that control expression of given small noncoding RNAs and describe their role in regulation of gene expression. Next, many examples of gene-specific regulatory effects in different cell types are described and summarized in the respective tables. Regulation became more complicated in stress conditions; it appears different for specific tissues and differentiation states. Again, many regulatory events were described in detail. That gives a broad perspective on the diversity of the regulatory effects and their significance in biogenesis of human diseases. The article is definitely worth to be published.

 I have some only some minor comments concerning the text and figures

  • The article describes regulation by small noncoding RNAs, but “small” is omitted in the title.

Reply – “small RNAs” is now included in the title.

  • Line 29 in the Introduction: “majority of genomic DNA is transcribed” instead “majority of all nucleotides is transcribed”.

Reply – Corrected as suggested. Line 30-31.

  • Lines 39-41: Please correct to avoid the mistakes: rRNA is not small RNA, tRNA is not constitutively expressed.

Reply – Corrected as suggested. Lines 43-44.

  • Line 47: “Regulatory small noncoding RNAs….”

Reply – Corrected as suggested. Line 53.

  • Table 1: are there examples of miRNA-mediated regulation in human cell?

Reply –Table legend changed to “Examples of miRNA-mediated transcription regulation in human cells.”

  • Line 378: Backsplicing process is not explained, and no reference is provided. Possibly the extended legend to Figure 3 would help in understanding what is going on.

Reply – Legend to figure 3 extended to describe the backsplicing process. The process was also described in lines 393-395.

Reference:

  1. Wang, Y.; Wang, Z. Efficient Backsplicing Produces Translatable Circular MRNAs. RNA 2015, 21, 172–179, doi:10.1261/RNA.048272.114.
  • Figure 4 should be more compact.

Reply – Figure 4 was edited to be more compact and replaced in the manuscript.

Author Response

Review 3

This is a good review. Solid, factual, informative, readable.

A couple of suggestions:

  • Since snoRNAs, snRNAs, tRNAs and derived fragments can also have regulatory functions, it would be good to indicate this in Figure 1 by way of dotted lines.

Reply – Corrected as suggested. Dotted lines added in Figure 1 to indicate the regulatory functions of tRNAs, snoRNAs, and snRNAs.

  • It might also be worth pointing out that some protein-coding loci also generate miRNAs or long noncoding RNAs by alternative splicing[1-10] and there is “a non-negligible fraction of protein coding genes (where) the major transcript does not code a protein”[11]. For example, the PNUTS gene encodes both PNUTS mRNA and lncRNAPNUTS by alternative splicing of the primary transcript, each eliciting distinct biological functions; PNUTS mRNA is ubiquitously expressed, whereas the production of lncRNAPNUTS is tightly regulated[1].

Reply – Corrected as suggested. Lines 32-35.